# Creation of New Oregano Genotypes with Different Terpene Chemotypes via Inter- and Intraspecific Hybridization

**DOI:** 10.3390/ijms24087320

**Published:** 2023-04-15

**Authors:** Meiyu Sun, Ningning Liu, Jiahui Miao, Yanan Zhang, Yuanpeng Hao, Jinzheng Zhang, Hui Li, Hongtong Bai, Lei Shi

**Affiliations:** 1Key Laboratory of Plant Resources, Institute of Botany, Chinese Academy of Sciences, Beijing 100093, China; 2China National Botanical Garden, Beijing 100093, China; 3University of Chinese Academy of Sciences, Beijing 100049, China

**Keywords:** oregano, essential oil, terpene chemotypes, hybrid, F_1_ lines, SSR molecular marker

## Abstract

Oregano is a medicinal and aromatic plant of value in the pharmaceutical, food, feed additive, and cosmetic industries. Oregano breeding is still in its infancy compared with traditional crops. In this study, we evaluated the phenotypes of 12 oregano genotypes and generated F_1_ progenies by hybridization. The density of leaf glandular secretory trichomes and the essential oil yield in the 12 oregano genotypes varied from 97–1017 per cm^2^ and 0.17–1.67%, respectively. These genotypes were divided into four terpene chemotypes: carvacrol-, thymol-, germacrene D/β-caryophyllene-, and linalool/β-ocimene-type. Based on phenotypic data and considering terpene chemotypes as the main breeding goal, six oregano hybrid combinations were performed. Simple sequence repeat (SSR) markers were developed based on unpublished whole-genome sequencing data of *Origanum vulgare*, and 64 codominant SSR primers were screened on the parents of the six oregano combinations. These codominant primers were used to determine the authenticity of 40 F_1_ lines, and 37 true hybrids were identified. These 37 F_1_ lines were divided into six terpene chemotypes: sabinene-, β-ocimene-, γ-terpinene-, thymol-, carvacrol-, and *p*-cymene-type, four of which (sabinene-, β-ocimene-, γ-terpinene-, and *p*-cymene-type) were novel (i.e., different from the chemotypes of parents). The terpene contents of 18 of the 37 F_1_ lines were higher than those of their parents. The above results lay a strong foundation for the creating of new germplasm resources, constructing of genetic linkage map, and mapping quantitative trait loci (QTLs) of key horticultural traits, and provide insights into the mechanism of terpenoid biosynthesis in oregano.

## 1. Introduction

Oregano is a perennial herb or semishrub of the genus *Origanum* and family Lamiaceae, which is mainly distributed and cultivated in the Mediterranean region, India, North America, and Mexico [1]. Only one *Origanum* species (*O. vulgare*) is grown in China, specifically in northwest China, north China, and south of the Yangtze River Basin. The main species and subspecies of oregano include *O. vulgare*, *O. laevigatum*, *O. onites*, *O. compactum*, *O. majorana*, *O. vulgare* subsp. *hirtum*, and *O. vulgare* subsp. *virens*, etc. [2]. Oregano, also known as pizza grass, is widely used as a spice in southern Europe and the Americas to enhance the flavor of foods. The whole plant of oregano is used as a raw material for the extraction of essential oil, which is added to perfumes for bathing and sauna. Oregano has antibacterial, antioxidant, antiviral, anti-inflammatory, analgesic, and immune regulatory properties, and has, therefore, been used in traditional medicine to treat sunstroke, fever, acute gastroenteritis, dysentery, and other diseases [3,4]. Pure oregano essential oil is natural and has no side effects, which is an environmentally friendly and safe substitute for food additives [5], chemical preservatives [6], and feed additives [7,8,9]. Oregano essential oil has also been recognized worldwide as a natural feed additive and has certain potential for the preservation of chicken [7], fish [8], and rabbit [9]. In conclusion, oregano is an economically important plant with great potential for functional development. The excellent varieties should be bred to promote the sustainable uses of oregano.

Oregano essential oil is mainly composed of terpenes, including monoterpenes and sesquiterpenes [10]. Monoterpenes include carvacrol, thymol, γ-terpinene, *p*-cymene, sabinene, sabinene hydrate, linalool, 1,8-cineol, and β-ocimene, etc. [10,11]. Sesquiterpenes include β-caryophyllene, germacrene D, β-bisabolene, and caryophyllene oxide, etc. [12]. The primary terpenes in most European oregano accessions are carvacrol and thymol, which are isomers and involved in a wide range of biological activities. Carvacrol and thymol exhibit antibacterial, antioxidant, antiviral, antiseptic, insect repellent, anticancer and antispasmodic properties, and can be used as antibiotics and insecticides [13,14,15,16]. β-Caryophyllene can be used as a local anesthetic and for other important pharmacological activities [17]. Germacrene D is an important flavor compound that can be used as an antibiotic, insecticide, and insect attractant [18]. To date, considerable work has been carried out on the analysis of oregano essential oil, and a variety of volatile terpenes have been identified. The essential oils of *O. majoricum*, *O. vulgare* ssp. *hirtum*, *O. onites*, *O. vulgare* subsp. *viridulum*, *O. vulgare*, *O. vulgare* ‘Hot & Spicy,’ and *O. × majorana* ‘Hippokrates’ are mainly composed of carvacrol (29.00–93.58%), thymol (14.31–60.14%), γ-terpinene (6.40–24.14%), *p*-cymene (2.16–51.3%), sabinene (5.50–27.8%), terpineol (8.00–25.05%), β-caryophyllene (1.08–25.27%), germacrene D (1.31–11.26%), and elixene (13.69–25.27%) [10,19,20,21,22]. Our main breeding objective is to effectively increase the contents of these terpenes through hybridization.

So far, little research has been undertaken on oregano breeding. Franz and Novak [23] identified essential oil yield, high carvacrol content, biomass, plant growth habit, and biotic and abiotic stress tolerance as the main targets of oregano breeding. Two new *O. vulgare* varieties, ‘Pierre’ and ‘Eli,’ were developed through multiyear and multisite assessments. The ‘Eli’ essential oil contained 72.77% carvacrol which was much higher than the commercial varieties in the same cultivation conditions [24]. American Jianming Company developed two new oregano varieties, ‘KI-Ov1750’ and ‘KI-Ov1850’. Through conventional breeding techniques, carvacrol content (5.14/100 g of dried leaves in ‘KI-Ov1750’) and thymol content (3.77/100 g of dried leaves in ‘KI-Ov1850’) were significantly higher than their parents [25]. Sarrou et al. [26] evaluated the essential oil yield, composition, and carvacrol content of *O. vulgare* ssp. *hirtum* population which was derived by self-pollination or open pollination, and selected individuals with high carvacrol content (81–86%). Oregano essential oil has great value in international trade and is an important source of pharmacologically active substances. Therefore, the development of varieties with high essential oil yield is one of the main objectives of oregano breeding program [27]. Carvacrol and thymol are the characteristic volatiles exhibiting a variety of biological activities [28,29,30], and are, therefore, considered important targets of oregano breeding. To achieve the objectives of oregano breeding, different oregano genotypes should be crossed, and new varieties with good performance should be selected.

In recent years, DNA markers have been widely used to analyze and identify medicinal plants [31]. By detecting the molecular markers closely linked to the target gene, individual plants containing the target gene can be screened. Compared with conventional breeding, molecular markers offer the advantages of high accuracy and quick results. Currently, the most commonly used molecular markers in plant studies include simple sequence repeat (SSR), random amplified polymorphic DNA (RAPD), restriction fragment length polymorphism (RFLP), sequence-related amplified polymorphism (SRAP), amplified fragment length polymorphism (AFLP), DNA barcoding, and single nucleotide polymorphism (SNP) [32,33]. Given the recent advances in molecular marker technology, it is increasingly being used to perform genetic diversity analysis, genetic map construction, parental analysis, and molecular breeding [34]. Wild relatives of plant species are valuable genetic resources and genetic diversity studies are analyzed in favor of resource conservation and sustainable use [35,36]. In the breeding of medicinal and aromatic plants, materials with large phenotypic differences and high essential oil yield are usually selected for breeding. Molecular genetic analysis has been used to distinguish among the different oregano species and to study intra- and interspecific diversity [37,38,39,40,41]. We aim to develop SSR markers based on the oregano genome sequence to lay a foundation for subsequent molecular breeding.

In this study, through the evaluation of 12 oregano genotypes, we aimed to develop new chemotypes of oregano, which conclude high target terpenes in essential oil. Through increasing the contents of principal terpenes in essential oil, many hybrids will be developed by crossing different species, subspecies, and varieties of oregano. We will take into account the important traits, such as essential oil yield, glandular secretory trichome (GST) density, plant type (growth habit), and leaf and flower morphology. We speculate that our results will facilitate the breeding of new chemotypes or germplasm resources with higher terpene contents, which will lay a foundation for the breeding of new oregano varieties. Six crosses were performed using different oregano genotypes (*O. × majorana* ‘Hippokrates’ [Omh] × *O. vulgare* [Ov], *O. vulgare* ‘Varona’ [Ovv] × *O. vulgare* subsp. *hirtum* [Ovh], Omh × *O. vulgare* ‘Creticum’ [Ovc], *O. vulgare* ‘Hot & Spicy’ [Ovhs] × *O. vulgare* ‘Samothrakb’ [Ovs], Ovv × Ovs, and Omh × Ovs), and F_1_ progenies were obtained. SSR primers were developed based on the whole-genome sequence of *O. vulgare* (Sun et al., unpublished data) and used to verify the authenticity of 40 F_1_ individuals resulting from the six crosses. Consequently, 37 F_1_ lines were identified as true hybrids. This study will be very useful for the generation of new oregano varieties, the mechanism of terpenoid biosynthesis, molecular marker-assisted selection (MAS), construction of genetic linkage map, and mapping of quantitative trait loci (QTLs).

## 2. Results

### 2.1. Phenotypic Evaluation of 12 Oregano Genotypes and Construction of F_1_ Hybrids

Oregano has valuable medicinal and aromatic properties, with either erect or creeping growth habits. In this study, we evaluated the phenotypes of one wild Chinese and eleven European oregano accessions. All 12 oregano genotypes examined in this study were erect type (Figure 1a), but differed in leaf color, flower color, and biomass. The calyx of oregano is campanulate and is covered with glandular trichomes, while the corolla is longer and extends out of the calyx tube with four stamens. The flower phenotype of 12 oregano genotypes is shown in Figure 1b. Ovc, Ovc, and Olr have large flowers, purplish-red edges of calyx teeth, and deep pink corolla; Ovgs, Omh, and Ova produce pale pink flowers; Ovhs, Ovag, Ovs, Ovh, Ovt, and Ovv possess white flowers, but Ovs has pink anthers; Ov, Ovc, Olr, Ovs, and Ovh flowers possess both stigma and stamen which are male-fertile; and Ovgs, Omh, Ovhs, Ova, Ovag, Ovt, and Ovv have stigma but no stamen which are male-sterile.

GSTs are special structures that can secrete and store essential oils. The density of GSTs on leaves is an important factor affecting essential oil yield. Both the adaxial and abaxial surfaces of oregano leaves are covered with GSTs (Figure 1c). GST density on the adaxial surface of leaves was the highest in Ovs and Ovhs, intermediate in Ovh and Ovv, and lowest in Ov and Olr (Figure 1d). Similarly, GST density on the abaxial surface of leaves was the highest in Ovs and Ovhs; intermediate in Ovh, Ova, Omh, and Ovt; and lowest in Olr, Ovgs, and Ovag. The GST densities of Ovs, Ovhs, Ovh, and Omh (1017, 900, 761, and 439 per cm^2^, respectively) were higher than those of other genotypes.

The essential oil yield of the 12 oregano genotypes differed greatly (Figure 1e). It was highest in Omh (1.67%); intermediate in Ovhs (1.54%), Ovh (1.38%), and Ovs (1.36%); and lowest in Ov (0.17%). The essential oil yield of other oregano genotypes ranged from 0.33% to 0.65%. The correlation analysis showed that the essential oil yield was positively correlated with the leaf GST density. The color of essential oil also showed some variation among the different oregano genotypes; for example, brownish-yellow in Olr, milky white and relatively clear in Omh, and golden in all other genotypes (Figure 1e and Appendix A).

The main components in the essential oils of 12 oregano genotypes were measured by gas chromatography-mass spectrometry (GC-MS), and 20 terpenoids with relative content > 0.20% were statistically analyzed (Table 1 and Figure 2a). In Ov essential oil, germacrene D and β-caryophyllene were the most abundant (22.01% and 16.24%, respectively), followed by caryophyllene oxide (13.51%) and β-ocimene (7.11%). In Ova essential oil, linalool showed the highest percentage (26.50%), followed by β-ocimene (16.81%), γ-terpinene (12.98%), and *p*-cymene (12.30%). In Olr essential oil, thymol was the most predominant terpene (22.95%), followed by carvacrol (13.80%) and γ-terpinene (9.58%). In Ovc essential oil, germacrene D and β-caryophyllene accounted for 16.68% and 12.55% of the volatile terpene content, respectively. In Ovv, carvacrol (22.28%) and γ-terpinene (13.53%) were the most abundant terpenes. In Ovt essential oil, linalool (23.18%) and β-ocimene (16.90%) were the most abundant, followed by *p*-cymene (11.68%) and γ-terpinene (11.59%). Thymol was the most abundant terpene in Ovgs (16.98%) and Omh (20.17%) essential oil. It was highest β-ocimene in Ovag (16.00%); thymol, carvacrol, and γ-terpinene in Ovs (32.89%, 19.08%, and 15.63%, respectively); carvacrol, γ-terpinene, and thymol in Ovh (35.59%, 16.75%, and 21.34%, respectively); and carvacrol in Ovhs (70.36%).

Next, cluster analysis was performed on the 12 oregano genotypes, based on 20 main terpenoids found in all genotypes. Results showed that the 12 oregano genotypes were grouped into four clusters (Figure 2b). The first cluster included Ovv, Ovh, and Ovhs; because carvacrol was the main terpenoid in all three genotypes, these genotypes were defined as carvacrol-type. Olr, Ovs, Omh, and Ovgs grouped into the second cluster, with thymol as the main terpenoid, and were defined as thymol-type. Cluster three comprised Ov and Ovc, which contained higher contents of both germacrene D and β-caryophyllene; therefore, the chemotype of these genotypes was defined as germacrene D/β-caryophyllene-type. Similarly, cluster four contained Ova, Ovt, and Ovag, with linalool/β-ocimene-type chemotype.

Principal component analysis (PCA) was performed on the main terpenoids found in the essential oil of all 12 oregano genotypes. The results of PCA are shown in Figure 2c (composition score diagram of oregano essential oils) and Figure 2d (factor loading diagram). PCA divided the 12 oregano genotypes into four groups, consistent with the results of cluster analysis. Principal component 1 (PC1) explained 58.25% of the variation in essential oil composition of the 12 genotypes, and PC2 explained 21.34% variation. The factor loading diagram (Figure 2d) showed that carvacrol, *p*-cymene, and 1-octen-3-ol were characteristic terpenoids of Ovv, Ovh, and Ovhs. Similarly, linalool and β-ocimene were the characteristic terpenoids of Ova, Ovt, and Ovag; β-caryophyllene and germacrene D were those of Ov and Ovc; and thymol and α-terpinene were those of Olr, Omh, Ovs, and Ovgs.

Different hybrid combinations were designed using male-sterile oregano genotypes (Ovgs, Omh, Ovhs, Ovv, and Ovt) as the female parent and male-fertile oregano genotypes (Ov, Ovc, Ovs, and Ovh) as the male parent. Finally, six hybrid combinations were selected for the authentication of F_1_ progenies (a total of 40 lines): Omh × Ov (7 lines), Ovv × Ovh (4 lines), Omh × Ovc (13 lines), Ovhs × Ovs (5 lines), Ovv × Ovs (7 lines), and Omh × Ovs (4 lines) (Table 2 and Appendix A).

### 2.2. Development and Application of SSRs in Six Hybrid Combinations

Genome and annotation using high-fidelity (HiFi) and chromatin conformation capture (Hi-C) technologies revealed that *O. vulgare* (Chinese wild oregano) contains 15 chromosomes, with a total length of 641.87 Mb (Sun et al., unpublished data). The *O. vulgare* genome was highly repetitive. The annotated repetitive sequences (460.18 Mb) accounted for 71.70% of the entire genome. Additionally, a total of 394,276 tandem repeats were identified, accounting for 3.49% (22.38 Mb) of the whole genome (Sun et al., unpublished data). A total of 166,943 SSR loci were detected, of which 157,564 (94.38%) could be used for primer design, while 9379 SSR loci (5.62%) could not be used for primer design. Among these ten contigs, Contig000064 showed the highest frequency of SSR loci (439 SSR loci per Mb), and Contig000024 showed the lowest frequency (231 SSR loci per Mb) (Figure 3a). Among the top ten contigs with the largest distribution of SSR sites, Contig000009 harbored the greatest number of SSR sites (5889), and Contig000093 showed the smallest number of SSR sites (3362) (Figure 3b). The majority of SSRs were trinucleotide repeats (12), followed by dinucleotide repeats (6) and tetranucleotide repeats (2) (Figure 3c). ATT/AAT accounted for 2.5% of the trinucleotide repeats, followed by TAA/TTA (2.1%), AGA/TCT (1.5%), ATA/TAT (1.5%), TTC/GAA (1.5%), AAG/CTT (1.3%), CCG/CGG (0.4%), CTC/GAG (0.4%), AGC/GCT (0.4%), CAG/CTG (0.3%), CGC/GCG (0.3%), and GCC/GGC (0.3%). Among the dinucleotide repeats, GA/TC was the most common (22.4%), followed by CT/AG (21.1%), TA/TA (14.3%), AT/AT (13.5%), TG/CA (4.4%), and GT/AC (4.1%). The two trinucleotide repeats were AAAT/ATTT (0.5%) and TTTA/TAAA (0.4%). The length of SSR motifs in *O. vulgare* genome ranged from 10 to 48 bp. As the SSR motif length increased, the proportion of SSR markers in *O. vulgare* genome gradually decreased. For example, SSR loci with a 10 bp motif length were the most abundant (54,593, 32.7%), followed by 12 bp motif SSRs (13.1%) and, finally, 38 bp motif SSRs (1076, 0.6%) (Figure 3d).

A total of 300 primer pairs were designed to verify the authenticity of F_1_ lines using SSR markers (Appendix A). To test these primer pairs, polymerase chain reaction (PCR) amplification was first carried out on the parents of the six hybrid combinations shown in Table 2. Analysis of SSR polymorphisms between the parents revealed that one or two bands were generally amplified in the parents. After many repetitions, SSR primer pairs showing good specificity and clear and reproducible bands were selected for the authentication of hybrid progenies (Figure 4). For example, the OvSSR176 marker showed codominance in the Omh × Ovc cross (Figure 4a) and amplified specific bands in the female parent Omh (arrow 1) and male parent Ovc (arrow 2). Therefore, this primer could be used to screen the F_1_ progeny of the Omh × Ovc cross. Similarly, OvSSR201 showed codominance in the Ovhs × Ovs cross (Figure 4b) and produced specific bands in the female parent Ovhs (arrow 3) and male parent Ovs (arrow 4); OvSSR278 showed codominance in the Ovv × Ovh cross (Figure 4c; arrow 5 in the female parent Ovv and arrow 6 in the male parent Ovh). According to this method, 20 primer pairs were selected for screening the progeny of combination 1 (Omh × Ov; Appendix A), 23 pairs for combination 2 (Ovv × Ovh; Appendix A), 17 pairs for combination 3 (Omh × Ovc; Appendix A), 20 pairs for combination 4 (Ovhs × Ovs; Appendix A), 20 pairs for combination 5 (Ovv × Ovs; Appendix A), and 16 pairs for combination 6 (Omh × Ovs; Appendix A).

During genotyping, lines showing the bands of both parents or only one parent are true hybrids, while those showing non-parental bands are pseudohybrids. In combination 1 (Omh × Ov), for example, OvSSR083 amplified a specific band (arrow 1) from the female parent Omh and a specific band (arrow 2) from the male parent Ov. Among the seven hybrids, F_1_ lines 1, 2, 3, and 7 showed the amplification of bands from both parents. Genotyping using the OvSSR126 marker also revealed complementary bands in F_1_ lines 4 and 6. These results indicate that all six F_1_ lines are true hybrids. Moreover, nonparental bands appeared in the amplification results of F_1_ lines 4 and 6, indicating the appearance of new SSR loci and the abundance of genetic variation. However, F_1_ line 5 showed only the maternal band when screened with all 20 pairs of SSR primers, which suggests that line 5 is a pseudohybrid (Figure 5a). In combination 2 (Ovv × Ovh), specific bands were amplified from the female parent Ovv (arrow 3) and male parent Ovh (arrow 4). Among the four Ovv × Ovh hybrids, F_1_ lines 1, 2, and 3 showed both maternal and paternal specific bands, indicating that these lines were true hybrids; however, F_1_ line 4 was a pseudohybrid since it produced no paternal band with any of the 23 SSR primer pairs (Figure 5b). Overall, among the 40 F_1_ lines examined using SSR markers, 37 were true hybrids, and 3 were pseudohybrids (Figure 5).

### 2.3. Identification of VOCs in the Leaves of Seven Parental Genotypes and 37 F_1_ Lines

A total of 37 true hybrids (F_1_-1 to F_1_-37) were identified from the progenies of six hybrid combinations including combination 1 (Omh × Ov), combination 2 (Ovv × Ovh), combination 3 (Omh × Ovc), combination 4 (Ovhs × Ovs), combination 5 (Ovv × Ovs), and combination 6 (Omh × Ovs). Because of time and plant growth cycle related limitations, VOCs in leaves were identified by headspace-solid phase microextraction (HS-SPME), and the plant type, flower type, and flower color of the 37 F_1_ hybrids were observed (Appendix A). The leaf VOCs of seven parental genotypes and 37 F_1_ hybrids are shown in Figure 6a and Appendix A.

#### 2.3.1. Combination 1 (Omh × Ov; Six Hybrids, F_1_-1 to F_1_-6)

Sabinene hydrate (20.17%) and thymol (16.71%) were the most abundant terpenoids in the female parent Omh, and germacrene D (22.01%) and β-caryophyllene (21.62%) were higher in male parent Ov. In F_1_-1 and F_1_-5, sabinene hydrate was the most abundant terpenoid (27.93% and 18.96%, respectively), followed by germacrene D (14.87% and 16.68%, respectively) and β-caryophyllene (13.36% and 10.40%, respectively), which was a collection of the main terpenoids found in the parents. In F_1_-3 and F_1_-6, γ-terpinene was the most abundant terpenoid (28.60% and 35.79%, respectively), followed by *p*-cymene (17.32% and 21.56%, respectively). The main terpenoids in F_1_-2 were β-ocimene (25.30%) and β-caryophyllene (19.50%), while the main terpenoids in F_1_-4 were γ-terpinene (23.80%) and sabinene hydrate (18.23%).

#### 2.3.2. Combination 2 (Ovv × Ovh; Three Hybrids, F_1_-7 to F_1_-9)

The most terpenoids in the female parent Ovv were carvacrol (22.28%), *p*-cymene (13.53%), and germacrene D (11.94%), and those in the male parent Ovh were thymol (21.34%), carvacrol (15.59%), and γ-terpinene (16.75%). In F_1_-7 and F_1_-8, the most abundant terpenoids were carvacrol (62.26% and 23.74%, respectively) and γ-terpinene (21.11% and 42.37%, respectively), and their contents were higher than those in their parents. The main terpenoids in F_1_-9 were γ-terpinene (46.30%), thymol (14.82%), carvacrol (14.15%), and *p*-cymene (12.02%), which was an aggregation of the important terpenoids found in the parents.

#### 2.3.3. Combination 3 (Omh × Ovc; 13 Hybrids, F_1_-10 to F_1_-22)

The most abundant terpenoids in the female parent Omh were sabinene hydrate (20.17%) and thymol (16.71%). The most abundant terpenoids in the male parent Ovc were sabinene (18.32%), β-ocimene (15.49%), β-caryophyllene (11.25%), and germacrene D (10.32%). The most abundant terpenoids in F_1_-10 were β-ocimene (27.05%), bicyclogermacrene (14.98%), sabinene (13.54%), and β-caryophyllene (12.66%); F_1_-11, F_1_-17, and F_1_-21 (sabinene chemotype): sabinene hydrate (15.73%, 15.82%, and 36.14%, respectively) and sabinene (6.74%, 6.20%, and 9.45%, respectively); F_1_-12 and F_1_-13 (γ-terpinene chemotype): γ-terpinene (25.74% and 23.55%, respectively); F_1_-14 and F_1_-15 (thymol chemotype): thymol (36.80% and 46.57%, respectively), *p*-cymene (14.78% and 19.56%, respectively), and γ-terpinene (12.72% and 13.41%, respectively); F_1_-16, F_1_-18, F_1_-20, and F_1_-22 (*p*-cymene chemotype): *p*-cymene (43.39%, 36.42%, 26.21%, and 33.44%, respectively); F_1_-19 (carvacrol chemotype): carvacrol (30.30%), *p*-cymene (23.41%), and β-caryophyllene (10.57%). Overall, different chemotypes appeared in combination 3, and the variation was great.

#### 2.3.4. Combination 4 (Ovhs × Ovs; Five Hybrids, F_1_-23 to F_1_-27)

The most abundant terpenoids in the female parent Ovhs were carvacrol (32.76%) and *p*-cymene (13.51%). The most abundant terpenoids in the male parent Ovs were thymol (14.93%), γ-terpinene (14.89%), and *p*-cymene (13.25%). The most abundant terpenoids in F_1_-23 to F_1_-27 (carvacrol chemotype) were carvacrol (31.48%, 50.05%, 52.65%, 63.52%, and 35.34%, respectively), γ-terpinene (21.14%, 8.00%, 18.74%, 11.30%, and 15.25%, respectively), and *p*-cymene (23.87%, 27.39%, 12.88%, 13.03%, and 17.43%, respectively). The carvacrol contents of F_1_-24, F_1_-25, and F_1_-26 were higher than those of their parents.

#### 2.3.5. Combination 5 (Ovv × Ovs; Seven Hybrids, F_1_-28 to F_1_-34)

Carvacrol (22.28%) and *p*-cymene (13.53%) were the most abundant terpenoids in the female parent Ovv, and thymol (14.93%), γ-terpinene (14.89%), and *p*-cymene (13.25%) were the most abundant terpenoids in the male parent Ovs. In F_1_-28 and F_1_-31 (γ-terpinene chemotype), the most abundant terpenoids were γ-terpinene (30.92% and 30.89%, respectively) and *p*-cymene (22.44% and 23.66%, respectively). The main terpenoids in F_1_-29, F_1_-30, and F_1_-32 were β-ocimene (39.63%, 20.19%, and 27.74%, respectively), germacrene D (11.06%, 14.83%, and 15.24%, respectively), and β-caryophyllene (9.23%, 13.74%, and 10.92%, respectively), and these hybrids were grouped into the β-ocimene chemotype cluster. The main terpenoids in F_1_-33 and F_1_-34 were carvacrol (52.66% and 26.47%, respectively), γ-terpinene (17.48% and 13.68%, respectively), and *p*-cymene (15.64% and 10.24%, respectively), and these hybrids were grouped into the carvacrol chemotype cluster; the carvacrol content of F_1_-33 and F_1_-34 was higher than that of the female parent Ovv.

#### 2.3.6. Combination 6 (Omh × Ovs; Three Hybrids, F_1_-35 to F_1_-37)

The most abundant terpenoids in the female parent Omh were sabinene hydrate (20.17%) and thymol (16.71%), and those in the male parent Ovs were thymol (14.93%), γ-terpinene (14.89%), and *p*-cymene (13.25%). The most abundant terpenoids were *p*-cymene (54.99%) in F_1_-35; *p*-cymene (25.14%), sabinene hydrate (22.30%), and carvacrol (16.34%) in F_1_-36; and sabinene hydrate (19.52%) and sabinene (15.20%) in F_1_-37. Thus, the hybrids contained the main terpenoids of both parents.

Overall, cluster analysis of the 37 hybrids, based on their VOC composition, revealed six chemotypes: sabinene-, β-ocimene-, γ-terpinene-, thymol-, carvacrol-, and *p*-cymene-type. Among these, sabinene-, β-ocimene-, γ-terpinene-, and *p*-cymene-type were new chemotypes, i.e., different from the chemotypes of the seven parents (Figure 6b).

## 3. Discussion

Among the 12 different oregano genotypes, Omh and Ovhs showed the highest essential oil yield (1.67% and 1.54%, respectively), followed by Ovh and Ovs (1.38% and 1.36%, respectively), while the essential oil yield of the remaining oregano genotypes varied from 0.17% to 0.65% (Figure 1e). In oregano, essential oil is synthesized in GSTs on the surface of stems, leaves, and flowers; thus, GSTs may be closely related to the yield and quality of essential oil [42]. Ovs, Ovhs, and Ovh showed relatively higher GST density on the adaxial and abaxial surfaces of leaves (1017, 900, and 761 per cm^2^, respectively) than the other genotypes (Figure 1d). We found that the higher the density of GSTs on leaves, the higher the essential oil yield of oregano. In addition, a certain correlation was observed between the yield of oregano essential oil and the extraction method. In this study, steam distillation was used to extract essential oil from whole oregano plants. However, essential oil could be extracted from the different parts of oregano plants (stem, leaf, and flower) separately to further compare and analyze its yield and terpene composition among the different organs. The yield of essential oil is also related to the harvest time, drying technique, and plant irrigation frequency [43,44]. The application of GC-MS and HS-SPME/GC-MS in this study was suitable for the determination of terpene compositions of oregano genotypes.

In this study, the 12 oregano genotypes showed variation in their essential oil contents and compositions (Figure 2a). For example, the main components of Omh essential oil were thymol (20.17%), terpinen-4-ol (14.90%), γ-terpinene (10.87%), and sabinene hydrate (10.71%). Ovh essential oil was mainly composed of carvacrol (35.59%), thymol (21.34%), and γ-terpinene (16.75%). The results showed that the dominant components in Ovh essential oil, carvacrol and thymol, were similar to those reported previously [22,27], but their proportions were slightly different. Similarly, the dominant components in Omh essential oil were similar to those reported previously [45], although their proportions were different. These differences may be related to environmental factors (such as temperature, humidity, and rainfall), harvest time, and location of plants [46]. This further proves that oregano is rich in genetic diversity, and its essential oil composition varies among the different varieties. In addition, essential oil extraction methods and plant growth environment also affect the composition and content of essential oil [47,48]. The thymol content of oregano essential oil determined in this study was slightly lower than that determined in previous reports, which may be used to set the distillation temperature. However, thymol and carvacrol are isomers, i.e., they can be reversibly converted into each other in plants under certain environmental conditions, which can also lead to differences in their contents in oregano essential oil [37].

The selection of parents is very important to achieve objectives of crossbreeding. To create excellent germplasm resources, materials with superior traits should be selected as parents. Essential oil yield and composition are important indicators of oregano quality. Understanding and mastering plant phenotypic diversity are of great significance for improving the utilization and creating of excellent germplasm resources. The focus of current research on oregano essential oil is on thymol and carvacrol, which are the main components of essential oil of most European oregano varieties [19,20,21,22]. However, the Chinese wild oregano used in this study contains the most abundant germacrene D and β-caryophyllene, which are reported to be important flavor compounds. Germacrene D exhibits antibiotic properties, repellent activity against insects and herbivores, insecticidal activity, and attractant or pheromone properties [18]. β-Caryophyllene, the main sesquiterpene in a variety of plant essential oils, is used as a fragrance in cosmetics and foods. It exhibits several important pharmacological properties including antioxidant, anti-inflammatory, anticancer, cardioprotective, hepatoprotective, antibacterial, immunomodulatory, and local anesthetic effects [17]. In this study, the chemotypes of European oregano accessions were mainly identified as thymol- and carvacrol-type, and these chemotypes are of important medicinal value. Therefore, oregano genotypes of different chemotypes were selected as parent materials to effectively aggregate these pharmacological properties into a single genetic background through crossbreeding.

Crossbreeding is an important approach for creating new germplasm resources because it can combine the favorable traits of parents into a single genetic background and enhance genetic diversity [35,36]. However, because of the long duration of the crossbreeding cycle, it is important to determine the authenticity of hybrids at an early stage. SSR markers are highly accurate, reproducible, and timesaving and are widely used to identify hybrid plants [31]. Therefore, crossbreeding and SSR markers can be applied together to improve the efficiency of germplasm development. In theory, a hybrid can be identified using one SSR primer pair [38,39,40,41]. However, in practice, the use of only one primer pair can lead to errors and have certain limitations. Therefore, to improve the identification accuracy, it is better to select multiple pairs of SSR primers for the simultaneous screening of a genotype. In this study, we observed the partial complementary bands of both parents and the generation of new bands in some F_1_ lines (Figure 5). It may be due to the high heterozygosity level of oregano, or the occurrence of homologous recombination or base mutation during meiosis. Thus, these markers could significantly improve the breeding efficiency of oregano by facilitating MAS, constructing F_2_ progeny-based genetic linkage maps, and QTL mapping of key horticultural traits.

The chemotypes and contents of terpenes in the essential oils of 12 oregano genotypes were evaluated (Figure 2). The results showed that seven parental genotypes could be divided into three chemotypes: carvacrol-, thymol-, and germacrene D/β-caryophyllene-type. Similarly, the 37 hybrids could be divided into six chemotypes: sabinene-, β-ocimene-, γ-terpinene-, thymol-, carvacrol-, and *p*-cymene-type. Among these, the sabinene-, β-ocimene-, γ-terpinene-, and *p*-cymene-type chemotypes of hybrids were new, i.e., different from the chemotypes of their parents. Thus, this study shows that the identification of oregano hybrids using SSR markers is feasible. The application of SSR markers could significantly improve the breeding efficiency of oregano by providing a foundation for MAS.

## 4. Materials and Methods

### 4.1. Plant Materials

One Chinese wild oregano species (*O. vulgare*) and eleven European oregano accessions (*O. vulgare* ‘Aurea’, *O. vulgare* ‘Aureum Gold’, *O. vulgare* ‘Creticum’, *O. vulgare* ‘Golden Shine’, *O. × majorana* ‘Hippokrates’, *O. vulgare* ‘Hot & Spicy,’ *O. laevigatum* ‘Rosenkuppel’, *O. vulgare* ‘Samothrakb’, *O. vulgare* ‘Thumble’s variety’, *O. vulgare* ‘Varona’, and *O. vulgare* subsp. *hirtum*) were used in this study. *O. vulgare* (NCBI: TaxID: 39174) was collected directly from its natural habitat in 2019. *O. vulgare* ‘Creticum’, *O. laevigatum* ‘Rosenkuppel’, and *O. × majorana* ‘Hippokrates’ were introduced from Germany (introduction no. 862-09, 516-2016, and 511-2016, respectively). *O. vulgare* ‘Aurea’, *O. vulgare* ‘Golden Shine’, *O. vulgare* ‘Hot & Spicy’, *O. vulgare* ‘Samothrakb’, *O. vulgare* ‘Varona’, and *O. vulgare* subsp. *hirtum* were introduced from the Czech Republic (introduction no. 505-2016, 711-2015, 634-2015, 2017032, 2017050, and 2018031, respectively). *O. vulgare* ‘Aureum Gold’ was introduced from Canada (introduction no. 2018004). *O. vulgare* ‘Thumble’s variety’ was introduced from Holland (introduction no. 866-2012). All oregano genotypes were grown in an experimental farm, Institute of Botany, Chinese Academy of Sciences (IB-CAS), Beijing, China.

### 4.2. Measurement of GST Density

GSTs were visualized using a stereomicroscope (Leica DVM6, Weztlar, Germany). The ImageJ software (http://rsb.info.nih.gov/ij) was used to count the number of GSTs and measure the leaf area. GST density was calculated based on three plants.

### 4.3. Extraction of Essential Oil

The essential oil from 12 oregano genotypes was isolated by steam distillation at 180–200 °C for 90 min. The essential oil yield (%) was calculated as volume (mL) of the isolated oil per 100 g of dried plant material. The isolated essential oil was dried using anhydrous sodium sulfate and stored at 4 °C until needed for further testing [49].

### 4.4. Analysis of Essential Oil by GC-MS

GC-MS analyses of 12 oregano essential oil samples was performed using Agilent 7890A-7000B gas chromatograph (Agilent, Palo Alto, CA, USA) equipped with Agilent 5975C MS detector (Agilent, USA). Volatiles were separated using the HP-5MS capillary column (30 m length, 250 m internal diameter [ID], 0.25 μm film thickness) and the following temperature program: 5 min at 60 °C, increase at the rate of 4 °C/min to 220 °C, increase at the rate of 60 °C/min to 250 °C, and hold at 250 °C for 5 min. The other parameters were as follows: injector and detector temperature, 250 °C; carrier gas, helium (He); flow rate, 1 m/min; split ratio, 1:10; acquisition range, 50–500 *m*/*z* in electron-impact mode; ionization voltage, 70 eV; and injected sample volume, 1 µL. The content of each compound (%) was determined based on the normalization of the GC peak areas. Identification of individual compounds was based on the comparison of retention indices (RIs), relative to a homologous series of n-alkanes (C7–C40), mass spectral data from the NIST library (v14.0), and data from scientific literature [50]. Retention indices were calculated using the following equation:$$RI=\left(100\times Z\right)+\left\{\left.100\times \frac{\left[RT\left(x\right)-RT\left(z\right)\right]}{\left[RT\left(z+1\right)-RT\left(z\right)\right]}\right\}\right.$$
where *RT*(*x*), *RT*(*z*), and *RT*(*z* + 1) for the retention time (RT) of the analyzed composition, previous n-alkane of analyzed composition and latter n-alkane of analyzed composition. *Z* for the number of carbons of the previous n-alkane of analyzed composition.

### 4.5. Hybridization Design

Considering essential oil composition and yield as the main breeding goals and taking into account other phenotypic traits, male-sterile genotypes (no or immature pollen) and male-fertile genotypes (with pollen) of oregano were used as the female parent and male parent, respectively. Different crosses were performed in 2020. Finally, six cross-combinations were selected for further analysis.

### 4.6. DNA Extraction

Leaves were collected from the plants of seven oregano accessions (Ovh, Ovv, Ovs, Ovhs, Ovc, Omh, and Ov) used as parents in six combinations (Omh × Ov, Ovv × Ovh, Omh × Ovc, Ovhs × Ovs, Ovv × Ovs, and Omh × Ovs), and 40 F_1_ lines, immediately frozen in liquid nitrogen, and stored at −80 °C. DNA was extracted from the leaves using DNA Secure Plant Kit (Tiangen, Beijing, China). The concentration and quality of the isolated DNA were assessed by electrophoresis on 1% agarose gel and using 2.0 Fluorometer (Life Technologies, Carlsbad, CA, USA).

### 4.7. Genotyping Using SSR Markers

All PCRs were performed on the PCR system (Bio-Rad, Hercules, CA, USA) in a 10 µL reaction volume containing 2 µL (20 ng/μL) of genomic DNA, 3 µL of 2 × Taq PCR Master Mix II (Tiangen, Beijing, China), 2 µL of forward and reverse primer mixture, and 3 µL of double distilled water (ddH_2_O). The thermocycling conditions were as follows: 94 °C for 3 min; 6 cycles of 94 °C for 45 s, 55–65 °C for 1 min, and 72 °C for 1 min; 9 cycles of 94 °C for 45 s, 50–58 °C for 1 min, and 72 °C for 1 min; 19 cycles of 94 °C for 30 s, 50 °C for 30 s, and 72 °C for 1 min; and final extension at 72 °C for 5 min. Amplification products were analyzed by electrophoresis on 8.0% (*w*/*v*) denaturing polyacrylamide gel in TBE buffer for 1 h using the DYY-6C electrophoresis apparatus (Beijing Liuyi Instrument Factory, Beijing, China) under a constant voltage of 220 V. DNA fragments were then visualized by silver staining (Silver sequence staining reagents, Promega, Madison, WI, USA) and sized with 50 bp DNA ladder (Tiangen, Beijing, China) [51]. SSR primer sequences are listed in Appendix A.

### 4.8. Analysis of Leaf VOCs by HS-SPME

The leaf VOCs of seven parental genotypes (Ovh, Ovv, Ovs, Ovhs, Ovc, Omh, and Ov) and 40 F_1_ lines were detected by HS-SPME. Briefly, 0.25 g of fresh leaf powder was weighed and immediately placed into a 20 mL headspace vial (Agilent, Palo Alto, CA, USA) containing 20 μL of internal standard solution (1 mg/mL 3-octanol, Cas#589-98-o; Aladdin, Shanghai, China). The vials were sealed using crimp-top caps with TFE-silicone headspace septa (Agilent, Palo Alto, CA, USA). Subsequently, each vial was immediately incubated at 40 °C for 30 min. Then, a 100 µm thick polydimethylsiloxane (PDMS)-coated fiber (Supelco, Inc., Bellefonte, PA, USA) was exposed to the headspace for 30 min to absorb the volatiles. All VOCs on the PDMS-coated fiber were then analyzed by GC-MS using Model 7890A GC instrument and 7000B mass spectrometer (Agilent, Palo Alto, CA, USA), as described previously [52], under the following conditions: injector and transfer line temperature, 250 °C and 250 °C, respectively; column temperature, 50 °C for 3 min, followed by a gradual increase to 150 °C at 4 °C/min for 2 min and a final increase to 250 °C at 8 °C/min for 5 min; carrier gas, He; flow rate, 1 mL/min; and injection mode, splitless. The identity of VOCs was determined by comparing their retention times with those of authentic standards. Agilent MassHunter 5.0 was used to analyze the chromatograms and mass spectra. VOCs were identified by comparing the retention times of individual peaks and identified the mass spectra using the mass spectra in the NIST database v14.0 and the literature [50]. Retention indices were calculated using the equation in Section 4.4.

### 4.9. Statistical Analysis

All samples were prepared and analyzed in triplicate, and data were expressed as mean ± standard deviation. All statistical analyses were performed using the SPSS software (version 23.0; SPSS, Chicago, IL, USA). PCA and other types of analyses were performed online (https://www.metaboanalyst.ca). PCA plots were evaluated using Unscrambler X (version 10.4). To further distinguish chemotypes of essential oils, a supervised statistical treatment was run using Origin (version 2021). Histogram of the relative contents of terpenoids was generated using GraphPad Prism (Version 8.3.0.538) [53].

## 5. Conclusions

Oregano terpenoid compositions have numerous applications in pharmaceutical, food, feed additive, and cosmetic industries, owing to their antioxidant, antibacterial, antifungal, anti-inflammatory, antiviral, and immunological properties. In this study, we evaluated the phenotypes of 12 oregano genotypes and generated F_1_ progenies by hybridization. SSR markers were developed based on unpublished whole-genome sequencing data of *Origanum vulgare*. These codominant primers were used to determine the authenticity of F_1_ lines. The chemotypes and contents of terpenes in the essential oils of seven parental genotypes could be divided into three chemotypes: carvacrol-, thymol-, and germacrene D/β-caryophyllene-type. Similarly, the F_1_ hybrids could be divided into six chemotypes: sabinene-, β-ocimene-, γ-terpinene-, thymol-, carvacrol-, and *p*-cymene-type. Among these, the sabinene-, β-ocimene-, γ-terpinene-, and *p*-cymene-type chemotypes of hybrids were new, i.e., different from the chemotypes of their parents. Thus, this study laid a strong foundation for the creation of new germplasm resources, the construction of a genetic linkage map, and mapping of QTLs of key horticultural traits, and provided insights into the mechanism of terpenoid biosynthesis in oregano.

## Figures and Tables

**Figure 1 ijms-24-07320-f001:**
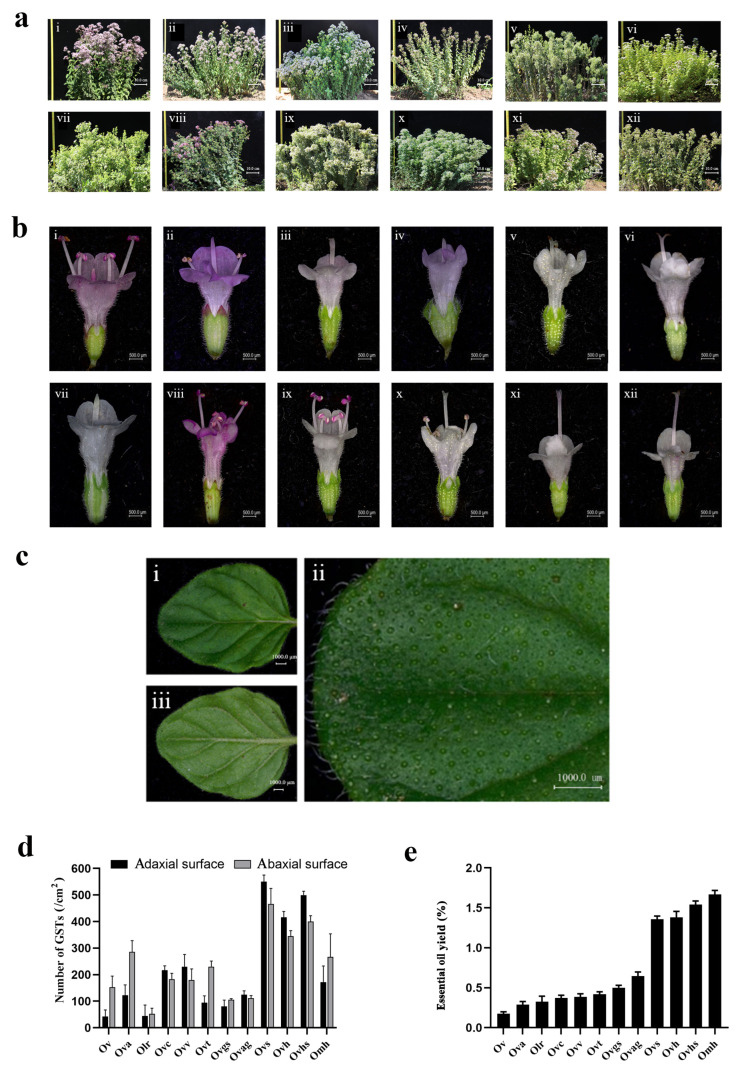
Phenotypic evaluation of one wild Chinese and eleven European oregano (*Origanum*) accessions. (**a**) Plant type. (**i**) *O. vulgare* (Ov); (**ii**) *O. vulgare* ‘Creticum’ (Ovc); (**iii**) *O. vulgare* ‘Golden Shine’ (Ovgs); (**iv**) *O. × majorana* ‘Hippokrates’ (Omh); (**v**) *O. vulgare* ‘Hot & Spicy’ (Ovhs); (**vi**) *O. vulgare* ‘Aurea’ (Ova); (**vii**) *O. vulgare* ‘Aureum Gold’ (Ovag); (**viii**) *O. laevigatum* ‘Rosenkuppel’ (Olr); (**ix**) *O. vulgare* ‘Samothrakb’ (Ovs); (**x**) *O. vulgare* subsp. *hirtum* (Ovh); (**xi**) *O. vulgare* ‘Thumble’s variety’ (Ovt); (**xii**) *O. vulgare* ‘Varona’ (Ovv). (**b**) Floral organ arrangement. Labels (**i**–**xii**) are the same as described in (**a**). Scale bars = 500.0 μm. (**c**) Glandular secretory trichomes (GSTs) on the adaxial surface (**i**) and abaxial surface (**ii**,**iii**) of leaves. Scale bars = 1000.0 μm. (**d**,**e**) GST density on the adaxial and abaxial surfaces of leaves (**d**) and essential oil yield (**e**) of 12 oregano genotypes.

**Figure 2 ijms-24-07320-f002:**
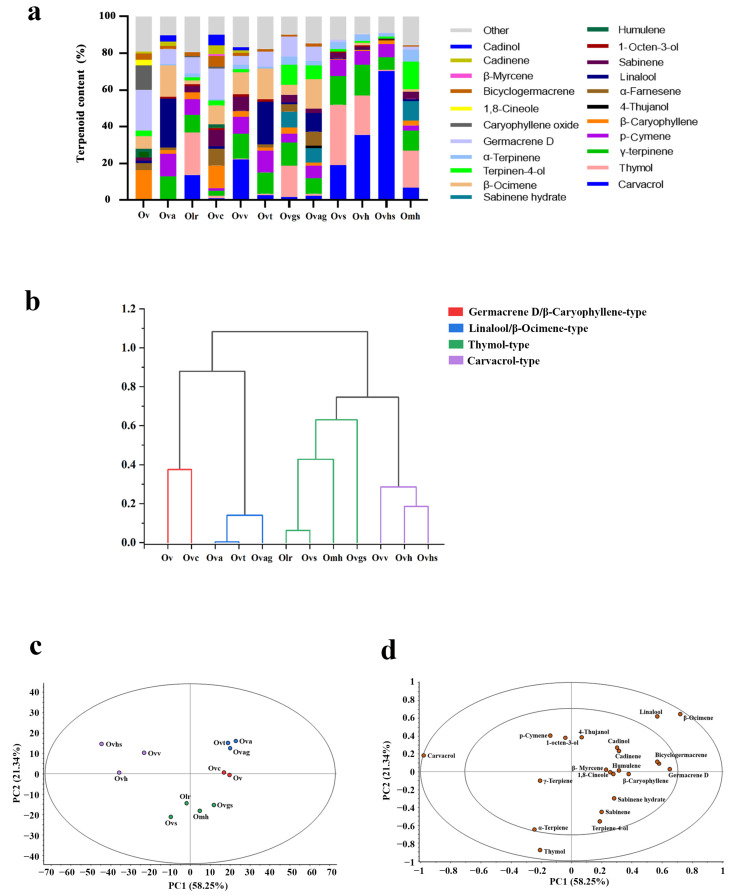
Relative content, cluster analysis, and principal component analysis (PCA) of volatile organic compounds (VOCs) in the essential oils of 12 oregano genotypes. (**a**) Histogram of the relative contents of terpenoids. (**b**) Cluster analysis of 12 oregano genotypes, based on their essential oil compositions. (**c**) Score plot of the tested oregano essential oils. (**d**) Loading plot of active terpenoids.

**Figure 3 ijms-24-07320-f003:**
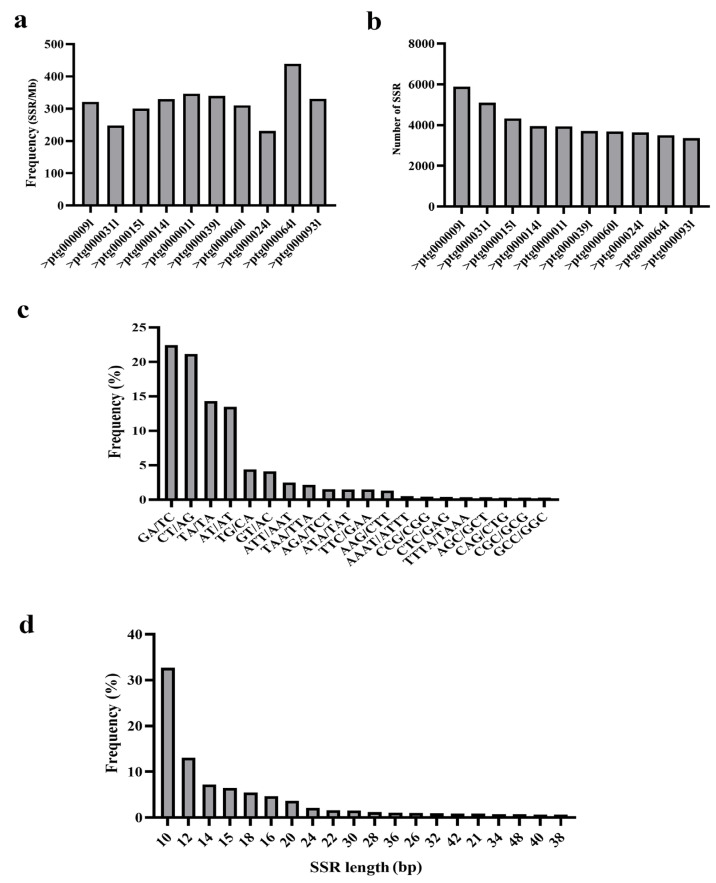
Detection of simple sequence repeat (SSR) loci in the *O. vulgare* genome. (**a**) Frequency of SSR loci (number of SSR loci per Mb) in the top ten longest contigs. (**b**) Total number of SSR loci on each of the ten longest contigs. (**c**) SSR motif type and frequency. (**d**) Frequency of SSR motifs of different lengths.

**Figure 4 ijms-24-07320-f004:**
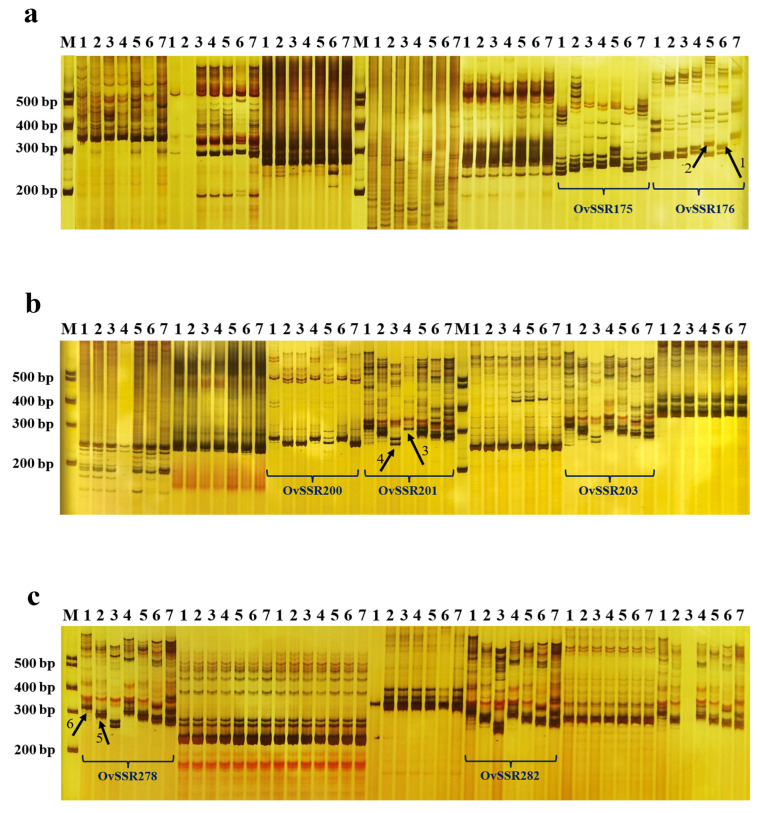
Genotyping of the seven parents of six hybrid combinations using codominant SSR markers. (**a**–**c**) Genotyping using OvSSR175 and OvSSR176 markers (**a**); OvSSR200, OvSSR201, and OvSSR203 markers (**b**); and OvSSR278 and OvSSR282 markers (**c**). M, DNA Marker; 1, Ovh; 2, Ovv; 3, Ovs; 4, Ovhs; 5, Ovc; 6, Omh; 7, Ov. Arrows 1–6 show the characteristic bands amplified in the parents of the six hybrid combinations.

**Figure 5 ijms-24-07320-f005:**
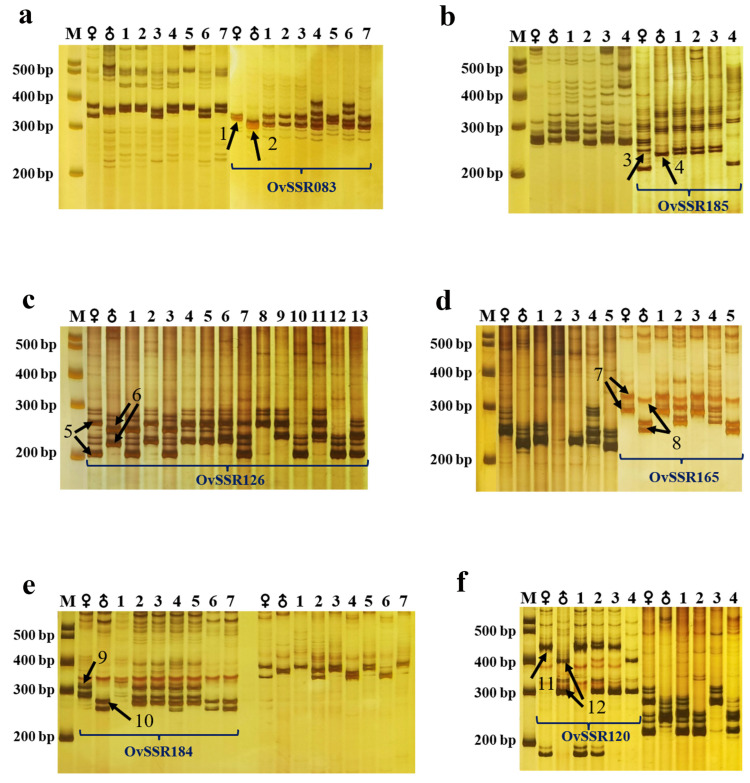
Genotyping of the F_1_ progeny of six hybrid combinations using codominant SSR markers. (**a**–**f**) Genotyping F_1_ lines of combination 1 (Omh × Ov) using the OvSSR083 marker (**a**), combination 2 (Ovv × Ovh) using the OvSSR185 marker (**b**), combination 3 (Omh × Ovc) using the OvSSR126 marker (**c**), combination 4 (Ovhs × Ovs) using the OvSSR165 marker (**d**), combination 5 (Ovv × Ovs) using the OvSSR184 marker (**e**), and combination 6 (Omh × Ovs) using the OvSSR120 marker (**f**). M, DNA Marker; **♀**, female parent (indicated on the left in each combination); **♂**, male parent (indicated on the right in each combination). Numbers represent F_1_ lines. Arrows 1, 3, 5, 7, 9, and 11 indicate characteristic bands of the female parents. Arrows 2, 4, 6, 8, 10, and 12 represent characteristic bands of the male parents.

**Figure 6 ijms-24-07320-f006:**
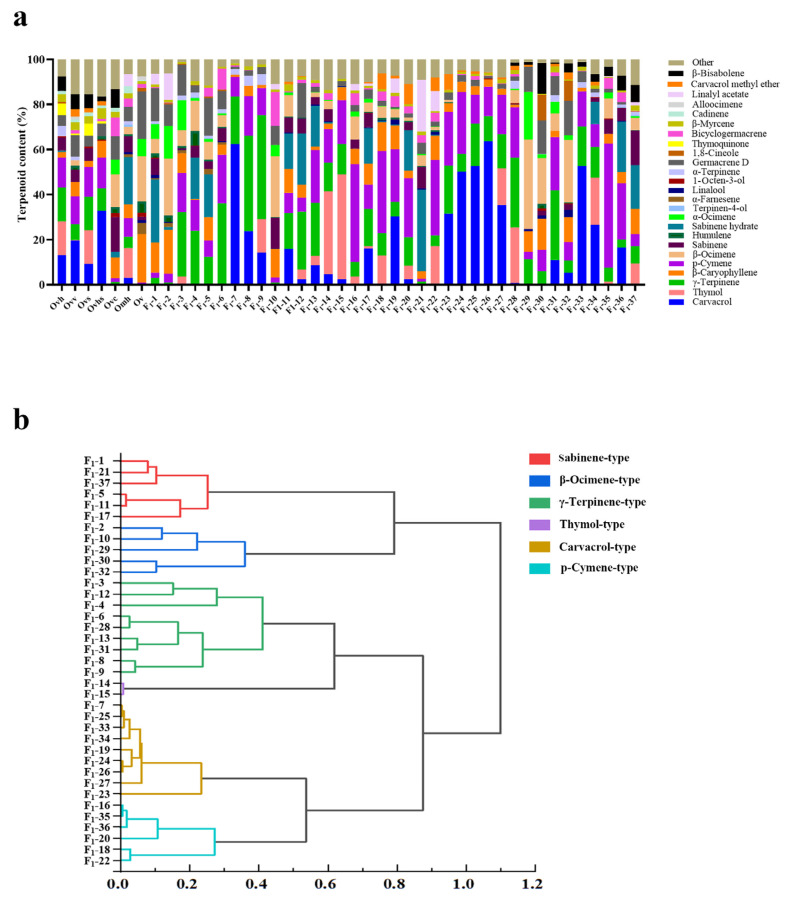
Analysis of volatile terpenoids in the leaves of seven parents and 37 F_1_ lines of six hybrid combinations. (**a**) Histogram of volatile terpenoids in the leaves of parental genotypes (Ovh, Ovv, Ovs, Ovhs, Ovc, Omh, and Ov) and 37 F_1_ lines. (**b**) Cluster analysis of 37 F_1_ lines, based on their terpenoid contents.

**Table 1 ijms-24-07320-t001:** Relative contents of volatile terpenoids in the essential oil of 12 oregano genotypes.

No.	Terpenoid	RI Cal ^1^	RI Lit ^2^	Relative Content (%) ^3^											
				Ov	Ova	Olr	Ovc	Ovv	Ovt	Ovgs	Ovag	Ovs	Ovh	Ovhs	Omh
**1**	Carvacrol	1300	1299	-	-	13.80 ± 0.04 e	1.11 ± 0.22 j	22.28 ± 0.38 c	2.60 ± 0.01 g	1.69 ± 0.15 i	2.43 ± 0.01 g	19.08 ± 0.25 d	35.59 ± 0.89 b	70.36 ± 0.49 a	6.80 ± 0.04 f
**2**	Thymol	1291	1291	-	-	22.95 ± 0.19 b	1.26 ± 0.25 f	0.36 ± 0.01 f	0.95 ± 0.01 f	16.98 ± 0.47 e	1.15 ± 0.02 f	32.89 ± 0.62 a	21.34 ± 0.24 c	0.55 ± 0.01 f	20.17 ± 2.08 d
**3**	γ-Terpinene	1057	1060	-	12.98 ± 0.09 c	9.58 ± 0.02 e	2.60 ± 0.50 h	13.53 ± 0.30 c	11.59 ± 0.01 d	12.75 ± 1.88 c	8.38 ± 0.06 f	15.63 ± 0.27 b	16.75 ± 0.08 a	6.94 ± 0.10 g	10.87 ± 0.06 d
**4**	*p*-Cymene	1023	1025	-	12.30 ± 0.03 a	8.68 ± 0.10 d	1.36 ± 0.26 j	9.25 ± 0.06 c	11.68 ± 0.04 b	4.79 ± 0.43 h	6.71 ± 0.02 g	8.65 ± 0.16 d	7.45 ± 0.19 e	7.02 ± 0.05 f	2.84 ± 0.01 i
**5**	β-Caryophyllene	1417	1419	16.24 ± 1.68 a	1.91 ± 0.02 c	3.53 ± 0.04 c	12.55 ± 0.12 b	3.16 ± 0.02 c	1.84 ± 0.01 c	3.32 ± 0.30 c	1.88 ± 0.01 c	0.70 ± 0.01 c	0.70 ± 0.01 c	2.00 ± 0.08 c	2.52 ± 0.01 c
**6**	Sabinene hydrate	1097	1077	-	-	-	-	-	-	8.67 ± 0.23 b	7.83 ± 0.07 b	-	-	-	10.71 ± 1.78 a
**7**	α-Farnesene	1508	1508	3.76 ± 0.08 c	1.41 ± 0.02 d	-	9.06 ± 0.75 a	-	1.58 ± 0.01 d	3.90 ± 0.34 c	7.70 ± 0.59 b	-	-	-	-
**8**	Linalool	1099	1099	1.40 ± 0.25 d	26.50 ± 0.26 a	-	1.35 ± 0.28 d	-	23.18 ± 0.17 b	1.17 ± 0.10 d	10.00 ± 1.32 c	0.33 ± 0.02 e	0.38 ± 0.02 e	-	1.12 ± 0.01 d
**9**	Sabinene	973	974	1.61 ± 0.16 g	-	3.44 ± 0.10 d e	8.88 ± 0.73 a	7.72 ± 0.01 b	-	4.19 ± 0.37 c	2.42 ± 0.01 f	3.04 ± 1.46 e f	1.46 ± 0.11 g	-	3.91 ± 0.03 c d
**10**	1-Octen-3-ol	978	980	-	1.29 ± 0.01 b c	1.05 ± 0.04 d	1.20 ± 0.27 c	1.42 ± 0.02 a b	1.45 ± 0.01 a	-	-	0.75 ± 0.02 e	0.53 ± 0.06 f	0.51 ± 0.01 f	-
**11**	Humulene	1451	1454	4.85 ± 1.04 a	-	-	1.96 ± 0.37 b	-	-	-	-	-	-	0.23 ± 0.01 c	-
**12**	β-Ocimene	1038	1037	7.11 ± 1.02 ^d^	16.81 ± 0.01 ^a^	2.20 ± 0.06 ^f^	10.32 ± 0.50 ^c^	11.94 ± 0.11 ^b^	16.90 ± 0.02 ^a^	5.15 ± 1.75 ^e^	16.00 ± 0.05 ^a^	-	1.54 ± 0.12 ^f^	-	1.45 ± 0.01 ^f^
**13**	Terpinen-4-ol	1175	1177	2.94 ± 1.03 ^d^	-	1.59 ± 0.04 ^e^	2.48 ± 0.45 ^d^	1.67 ± 0.01 ^e^	-	11.10 ± 0.98 ^b^	7.64 ± 0.04 ^c^	1.25 ± 0.03 ^e^	0.93 ± 0.09 ^e^	1.04 ± 0.02 ^e^	14.90 ± 0.03 ^a^
**14**	α-Terpinene	1015	1017	-	0.83 ± 0.03 ^h^	2.13 ± 0.08 ^f^	0.72 ± 0.13 ^h^	2.47 ± 0.03 ^e^	0.95 ± 0.01 ^h^	4.49 ± 0.40 ^b^	2.29 ± 0.02 ^e f^	3.67 ± 0.08 ^c^	3.25 ± 0.24 ^d^	1.77 ± 0.05 ^g^	6.56 ± 0.04 ^a^
**15**	Germacrene D	1480	1481	22.01 ± 2.95 ^a^	8.36 ± 0.03 ^d^	8.79 ± 0.03 ^d^	16.68 ± 0.08 ^b^	4.66 ± 0.03 ^e^	8.14 ± 0.03 ^d^	10.72 ± 0.95 ^c^	8.05 ± 0.17 ^d^	1.23 ± 0.05 ^f^	0.62 ± 0.06 ^f^	0.34 ± 0.10 ^f^	1.57 ± 0.06 ^f^
**16**	Caryophyllene oxide	1581	1581	13.51 ± 1.35 ^a^	-	0.72 ± 0.02 ^b c^	0.97 ± 0.18 ^b^	-	-	-	-	-	-	-	-
**17**	1,8-Cineole	1029	1032	2.92 ± 0.99 ^d^	-	-	-	-	-	-	-	-	-	-	-
**18**	Bicyclogermacrene	1495	1495	3.38 ± 0.09 ^b^	1.59 ± 0.02 ^c d^	1.98 ± 0.02 ^c^	5.90 ± 0.63 ^a^	1.77 ± 0.02 ^c d^	1.53 ± 0.01 ^c d^	1.32 ± 0.12 ^d^	1.77 ± 0.02 ^c d^	-	-	-	0.86 ± 0.03 ^e^
**19**	Cadinene	1522	1518	1.05 ± 0.02 ^c^	2.22 ± 0.01 ^b^	-	4.97 ± 1.50 ^a^	1.50 ± 0.06 ^b c^	-	-	-	-	-	-	-
**20**	Cadinol	1640	1640	-	3.44 ± 0.05 ^b^	-	5.80 ± 0.89 ^a^	1.40 ± 0.01 ^c^	-	-	-	-	-	-	-

^1^ RI Cal, Retention Index (RI) values calculated according to C7–C40. ^2^ RI Lit, RI values obtained by searching the mass spectral database NIST v14.0. ^3^ Ov, *O. vulgare*; Ova, *O. vulgare* ‘Aurea’; Olr, *O. laevigatum* ‘Rosenkuppel’; Ovc, *O. vulgare* ‘Creticum’; Ovv, *O. vulgare* ‘Varona’; Ovt, *O. vulgare* ‘Thumble’s variety’; Ovgs, *O. vulgare* ‘Golden Shine’; Ovag, *O. vulgare* ‘Aureum Gold’; Ovs, *O. vulgare* ‘Samothrakb’; Ovh, *O. vulgare* subsp. *hirtum*; Ovhs, *O. vulgare* ‘Hot & Spicy’; Omh, *O.* × *majorana* ‘Hippokrates’. Means with different letters in a row are statistically significant (*p* < 0.05).

**Table 2 ijms-24-07320-t002:** Characteristics of six oregano hybrid combinations.

Characteristic	Combination 1		Combination 2		Combination 3		Combination 4		Combination 5		Combination 6	
	♀ Omh	♂ Ov	♀ Ovv	♂ Ovh	♀ Omh	♂ Ovc	♀ Ovhs	♂ Ovs	♀ Ovv	♂ Ovs	♀ Omh	♂ Ovs
Chemotype	Thymol	β-caryophyllene	Carvacrol	Carvacrol	Thymol	Germacrene D	Carvacrol	Thymol	Carvacrol	Thymol	Thymol	Thymol
Essential oil yield (%)	1.62	0.15	0.37	1.30	1.62	0.33	1.50	1.40	0.37	1.40	1.62	1.40
GST density (per cm^2^)	439	194	411	761	439	400	900	1017	411	1017	439	1017
No. of lines	7		4		13		5		7		4	

♀, female parent; ♂, male parent.

## Data Availability

Raw whole-genome sequence data and gene sequences of *Origanum vulgare* have been deposited in the NCBI GenBank database under BioProject accession number PRJNA842006.

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
