# Peer review of "Creation of New Oregano Genotypes with Different Terpene Chemotypes via Inter- and Intraspecific Hybridization"

_ijms, 2023, doi:10.3390/ijms24087320_

Round 1

Reviewer 1 Report (Previous Reviewer 2)

In my opinion, the final version of the manuscript after resubmission is comprehensively improved and corrected. Thus, it shall be accepted for publication.

Author Response

Response: Thank you very much for your many great suggestions and comments last time.

Reviewer 2 Report (Previous Reviewer 3)

Manuscript ID: ijms-2292609

Title:      Creation of New Oregano Genotypes with Different Terpene Chemotypes via Inter- and Intraspecific Hybridization

Comments to the Author

This manuscript is focusing on one of the current global issues of metal pollution which detrimentally effects human diet and is a great health risks. The theme of the paper is evaluated the phenotypes of 12 oregano genotypes and generated F1 progenies by hybridization. Overall I think more attention should be devoted to MINOR REVISION before going for acceptance. There are some grammatical and typographic mistakes in the manuscript which needs to be addressed. The authors must use short sentences to avoid grammar and typo mistakes. My advice and suggestions are as follow

Comments:

Abstract

-Abstract is OK and clearly summarize the whole experiment, but I recommend detailing the purpose of this study

Introduction

-        Similar type of spelling and typographic mistakes have been seen throughout the text, make correction for each error. Please check the expression in English

Rezultatele si discutiile

They are properly described and organized. The statistical analysis seems to be properly done. The results are easy to follow. The discussions are detailed and appropriately detailed. But still lacks proof reading as it also has some spelling errors and space issues

Methods

They are properly described so that they can be replicated

Conclusion:

Conclusion was adequate and elaborates well

Author Response

Response: Thank you very much for your many great suggestions and comments. We have revised the grammar and typo mistakes, and used short sentences throughout the manuscript. In addition to these changes, the manuscript has been revised the language by two PhDs in related field.

Reviewer 3 Report (New Reviewer)

The authors present a good paper, to which the following considerations will be made:

Of the 53 articles contributed by the authors, more than half are from before 2018. The references must be as up-to-date as possible.

With what objective is it intended to create an oregano chemotype with a higher content of thymol and carvacrol or other monoterpenes, if the existing chemotypes already have antimicrobial, antioxidant, antiviral, anti-inflammatory, analgesic properties?

To determine if this new chemotype truly presents an improvement with respect to those that already exist, its properties must be tested, and its activity compared with the existing ones.

A study of antimicrobial, or antioxidant, or antiviral, or anti-inflammatory, or analgesic activity should be included.

Author Response

Response: Thank you very much for your advices. We have replaced 19 references with new as much as possible.

Our research team mainly conducts researches on the breeding, synthesis mechanism and function study of main components of aromatic plants such as lavender, thyme, oregano, and rosemary, etc. The published papers are as follows:

  1. Yanmei Dong, Wenying Zhang, Jingrui Li , Di Wang, Hongtong Bai , Hui Li*, Lei Shi*. The transcription factor LaMYC4 from lavender regulates volatile Terpenoid biosynthesis. BMC Plant Biology. 2022.22:289.
  2. Yanmei Dong, Jingrui Li, Wenying Zhang, Hongtong Bai, Hui Li*, Lei Shi*. Exogenous application of methyl jasmonate affects the emissions of volatile compounds in lavender (Lavandula angustifolia). Plant Physiology and Biochemistry. 2022.185:25-34.
  3. Jingrui Li#, Yiming Wang#, Yanmei Dong, Wenying Zhang, Di Wang, Hongtong Bai, Kui Li*, Hui Li*, Lei Shi*. The chromosome-based lavender genome provides new insights into Lamiaceae evolution and terpenoid biosynthesis. Horticulture Research. 2021.8:53
  4. Hui Li#, Jingrui Li#, Yanmei Dong, Haiping Hao, Zhengyi Ling, Hongtong Bai, Huafang Wang*, Hongxia Cui*, Lei Shi*. Time-series transcriptome provides insights into the gene regulation network involved in the volatile terpenoid metabolism during the flower development of lavender. BMC Plant Biology. 2019.19:313.
  5. Meiyu Sun#, Yanan Zhang#, Hongtong Bai, Guofeng Sun, Jinzheng Zhang*, Lei Shi*.Population diversity analyses provide insights into key horticultural traits of Chinese native thymes. Horticulture Research. 10:uhac262.
  6. Meiyu Sun#, Li Zhu#, Yanan Zhang, Ningning Liu, Jinzheng Zhang, Hui Li, Hongtong Bai, Lei Shi*. Creation of new germplasm resources, development of SSR markers, and screening of monoterpene synthases in thyme. BMC Plant Biology.23:13.
  7. Meiyu Sun#, Yanan Zhang#, Li Zhu, Ningning Liu,Hongtong Bai, Guofeng Sun, Jinzheng Zhang, Lei Shi*. Chromosome-level assembly and analysis of the Thymus genome provide insights on glandular secretory trichome formation and monoterpenoid biosynthesis in thyme. Plant Communications. 2022.3:100413.
  8. Yuanpeng Hao, Jiamu Kang, Xiaoqi Guo, Meiyu Sun, Hui Li, Hongtong Bai, Hongxia Cui, Lei Shi*. pH-responsive chitosan-based film containing oregano essential oil and black rice bran anthocyanin for preserving pork and monitoring freshness, Food Chemistry. 2023. 403:134393.
  9. Yuanpeng Hao, Jiamu Kang, Rui Yang, Hui Li, Hongxia Cui, Hongtong Bai, Andrey Tsitsilin, Jingyi Li, Lei Shi*. Multidimensional exploration of essential oils generated via eight oregano cultivars: compositions, chemodiversities, and antibacterial capacities. Food Chemistry. 2022.374:131629.
  10. Yuanpeng Hao, Xiaoqi Guo, Rui Yang, Yihao Yan, Meiyu Sun, Hui Li, Hongtong Bai, Hongxia Cui, Jingyi Li*, Lei Shi*. Unraveling the biosynthesis of carvacrol in different tissues of Origanum vulgare. International Journal of Molecular Sciences. 2022. 23:13231.
  11. Yuanpeng Hao, Xiaoqi Guo, Wenying Zhang, Fei Xia, Evan Yang, Hui Li, Hongtong Bai, Lei Shi*. Label-free quantitative proteomics reveals the antibacterial mechanism of rosemary essential oil against Salmonella entericaserovar Typhimurium. Industrial Crops & Products. 2022. 189:115757.
  12. Yuanpeng Hao, Jingyi Li, Lei Shi*. A Carvacrol-rich essential oil extracted from oregano (Origanum vulgare‘Hot & Spicy’) exerts potent antibacterial effects against Staphylococcus aureus. Frontiers in Microbiology. 2021.12:741861.
  13. Yuanpeng Hao, Jingyi Li, Wenying Zhang, Meiyu Sun, Hui Li, Fei Xia, Hongxia Cui, Hongtong Bai, Lei Shi*. Analysis of the chemical profiles and anti-aureusactivities of essential oils extracted from different parts of three oregano cultivars. Foods. 2021. 10: 2328.
  14. Xiaoqi Guo, Yuanpeng Hao, Wenying Zhang, Fei Xia, Hongtong Bai, Hui Li*, Lei Shi*. Comparison of origanum essential oil chemical compounds and their antibacterial activity against Cronobacter sakazakii. Molecules. 2022.27:6702.

Oregano, a material rarely bred, researched and utilized in China before, has been found by our research team to have strong antibacterial function of oregano essential oil, and we hope to cultivate, generate and utilize it on a large scale in China. But the vast majority of the species currently in China are wild and cultivated species introduced from Europe. There is only one wild species Origanum vulgare in China, and the essential oil components of which are mainly sesquiterpenes germacrene D and β-caryophyllene are determined. We are in urgent need of new oregano varieties with Chinese independent intellectual property rights for subsequent large-scale cultivation, generation, and utilization. The research of this paper is to create new germplasm, which will lay the foundation for breeding new varieties of oregano with Chinese independent intellectual property rights.

Only one plant with new chemotype is created in this manuscript, and we need tissue culture or propagation by cutting to increase the number of plants for functional study. At present, we have expanded the propagation of these new germplasms. In 2023, the application of new varieties in China and the functional study such as antimicrobial, antioxidant, antiviral, anti-inflammatory, or analgesic activity of new chemotypes of essential oils will be carried out. These are independent results, and we will continue to publish new papers.

This manuscript is a resubmission of an earlier submission. The following is a list of the peer review reports and author responses from that submission.

Round 1

Reviewer 1 Report

The manuscript (ijms-2190278) titled as “Creation of oregano new germplasms with different terpene chemotypes” Described the breeding practice to develop new germline with altered/novel terpene profile.

The major concern as a Reviewer I have with this study is that the authors have profiled the terpene content in F1 generation, where in breeding F1 lines are not ideal and most of the characteristics are losed/evaded/ruined in the next generation. As a reviewer I`ll ask the authors to go for F2 generation and perform the terpene profile.

The introduction of section of the article is too long and most of the content or informations are repeated in manuscript, better to give a concise intro of the study and highlight the objectives of the study.

In the Figure 2b, the data bars doesn`t show any statistical analysis data?

As a reviewer I am confused, why the authors were using so many (12) oregano species in this study? It is best to put the oregano species with silent features in the manuscript as main data and separate the remaining one as supplementary data. The length of the manuscript makes it difficult to pin point the major or minor issues individually.

I also I`ll suggest to send this manuscript for review to expert in breeding.

Thanks

Author Response

Response to Reviewers

Reviewer 1:

The manuscript (ijms-2190278) titled as “Creation of oregano new germplasms with different terpene chemotypes” Described the breeding practice to develop new germline with altered/novel terpene profile.

The major concern as a Reviewer I have with this study is that the authors have profiled the terpene content in F1 generation, where in breeding F1 lines are not ideal and most of the characteristics are losed/evaded/ruined in the next generation. As a reviewer I'll ask the authors to go for F2 generation and perform the terpene profile.

Response: It is likely that F1 lines are not ideal for analyzing the terpene contents in oregano essential oil since some of the characteristics may be lost in the next generation. However, we could not generate and analyze F2 plants for the current study because of limited time availability. Nonetheless, we aim to examine the terpene profile of oregano species in the near future and publish the results in the next article.

The introduction of section of the article is too long and most of the content or informations are repeated in manuscript, better to give a concise intro of the study and highlight the objectives of the study.

Response: We have revised the manuscript, including the Introduction, extensively and have tried to avoid the repetition of content. We believe that the revised version of this manuscript is more concise and clearly highlights the research objectives.

In the Figure 2b, the data bars doesn’t show any statistical analysis data?

Response: Figure 2b was drawn using Origin (version 2021) using 20 main terpenes (relative content >0.20%) found in the essential oil of all 12 oregano species. The oregano species were grouped into four clusters, based on their essential oil compositions, and were defined as having germacrene D/β-caryophyllene-, linalool/β-ocimene-, thymol-, and carvacrol-type chemotypes.

As a reviewer I am confused, why the authors were using so many (12) oregano species in this study? It is best to put the oregano species with silent features in the manuscript as main data and separate the remaining one as supplementary data. The length of the manuscript makes it difficult to pin point the major or minor issues individually.

Response: We first selected 12 different oregano species, subspecies, varieties, and hybrids with superior plant height and biomass among the collected oregano germplasm. Among these 12 oregano genotypes, O. vulgare and O. laevigatum are two important species, O. vulgare subsp. hirtum (Ovh) is an important subspecies of O. vulgareO. × majorana 'Hippokrates' (Omh) is an important hybrid between O. vulgare and O. majorana, and the remaining genotypes are oregano varieties with excellent phenotypic traits. Our breeding goals were to create new oregano hybrids with high yield of essential oil and high content of principal terpenes. Finally, we selected seven oregano genotypes (Omh, Ov, Ovv, Ovh, Ovc, Ovhs, and Ovs) with significant differences in oil yield and terpene contents for hybridization.

I also I`ll suggest to send this manuscript for review to expert in breeding.

Response: The International Journal of Molecular Sciences also sent our manuscript to plant breeding experts for review.

Reviewer 2 Report

Dear Authors,

The presented manuscript is an interesting work. However, some suggestions (written below) should be taken into consideration before publication:

Abstract. The amount of the essential oil should be expressed in the same unit as in the manuscript (% instead of mL/100g).

Introduction. This chapter is too long. It is recommended to shorten it, and order information, as following: general presentation of oregano; raw material and its chemical composition, biological activity, application in various branches of industry, breeding isuues. Then, brief purpose of the study, without metodology details.

Line 37, 38. O.vulgare ssp. vulgare and O.vulgare ssp. hirtum are subspecies not species.

Line 80-104. The description of EO extraction methods is not necessary here.

Results. Glandular trichomes are not the organs, ‘structures’ sounds better.

Figure 2. This figure should be improved graphically. It seems that Fig 1.a contains the same results as Table 1.

Discusion of the results is too long, please, shorten it visibly. In general, the results are valuable both from practice and scientific view point but their desription needs serious improvement and clarification to make it understandable for readers.

Materials and methods. Please, order the subchaptes in the same way as You present the results.

Comment for future investigations: it would be good to apply one separation techniques (traditional GC MS or SPME/GC MS) for all steps of the experiment.

Sincerely Yours,

Author Response

Response to Reviewers

Reviewer 2:

Dear Authors,

The presented manuscript is an interesting work. However, some suggestions (written below) should be taken into consideration before publication:

Abstract. The amount of the essential oil should be expressed in the same unit as in the manuscript (% instead of mL/100g).

Response: According to the reviewer’s suggestion, we have changed the unit of measurement of essential oil yield from mL/100g to % in the revised manuscript.

Introduction. This chapter is too long. It is recommended to shorten it, and order information, as following: general presentation of oregano; raw material and its chemical composition, biological activity, application in various branches of industry, breeding isuues. Then, brief purpose of the study, without metodology details.

Response: We have extensively revised the manuscript, and the objectives of the study and methodology details have been described.

Line 37, 38. O.vulgare ssp. vulgare and O.vulgare ssp. hirtum are subspecies not species.

Response: We have corrected this error in the revised manuscript.

Line 80-104. The description of EO extraction methods is not necessary here.

Results. Glandular trichomes are not the organs, ‘structures’ sounds better.

Response: We have deleted the description of EO extraction methods and have referred to GSTs as “structures” rather than as “organs” in the revised manuscript.

Figure 2. This figure should be improved graphically. It seems that Fig 1.a contains the same results as Table 1.

Response: We improved Figure 2 in the revised manuscript. Figure 2a is a graphical display of the data summarized in Table 1, i.e., 20 main terpenes (relative content > 0.20%) found in the essential oils of all 12 oregano genotypes.

Discusion of the results is too long, please, shorten it visibly. In general, the results are valuable both from practice and scientific view point but their desription needs serious improvement and clarification to make it understandable for readers.

Response: According to the reviewer’s suggestion, we have revised the manuscript, including the Discussion section, extensively. The Results section has also been heavily edited to improve clarity.

Materials and methods. Please, order the subchaptes in the same way as You present the results.

Response: To address the reviewer’s comment, subsections in Materials and Methods have been reordered to match the order of the results.

Comment for future investigations: it would be good to apply one separation techniques (traditional GC MS or SPME/GC MS) for all steps of the experiment.

Response: We will take the reviewer’s suggestion into consideration for future studies.

Reviewer 3 Report

The manuscript with the title "Creation of oregano new germplasms with different terpene chemotypes" addresses a topic of great interest. The creation of new germplasm is a current topic, pursued for several matrices.

Based on the phenotypic data, taking terpenic chemotypes as the main breeding objective, six combinations of oregano hybrids were constructed by crossing. The results of this study lay the foundation for the creation of new germplasm resources, genetic linkage map construction, quantitative trait locus (QTL) localization, and insight into the mechanism of terpenoid biosynthesis in oregano.

It is a very well written and organized sheet. It is an extensive and correspondingly detailed study. The introduction provides sufficient information about the existing base in specialized literature. The results are beautifully presented, with relevant evidence from each stage, clear pictures. The results are appropriately interpreted statistically both in tables and in figures. The discussions are relevant and well justified with specialized literature. The materials and methods are properly detailed to be able to put into practice the mentioned protocol.

Even if it is not mandatory, I recommend a detailed and well-structured conclusion, which provides an overview of this manuscript.

Author Response

Response to Reviewers

Reviewer 3:

The manuscript with the title "Creation of oregano new germplasms with different terpene chemotypes" addresses a topic of great interest. The creation of new germplasm is a current topic, pursued for several matrices.

Based on the phenotypic data, taking terpenic chemotypes as the main breeding objective, six combinations of oregano hybrids were constructed by crossing. The results of this study lay the foundation for the creation of new germplasm resources, genetic linkage map construction, quantitative trait locus (QTL) localization, and insight into the mechanism of terpenoid biosynthesis in oregano.

It is a very well written and organized sheet. It is an extensive and correspondingly detailed study. The introduction provides sufficient information about the existing base in specialized literature. The results are beautifully presented, with relevant evidence from each stage, clear pictures. The results are appropriately interpreted statistically both in tables and in figures. The discussions are relevant and well justified with specialized literature. The materials and methods are properly detailed to be able to put into practice the mentioned protocol.

Even if it is not mandatory, I recommend a detailed and well-structured conclusion, which provides an overview of this manuscript.

Response: We have revised the entire manuscript. Additionally, English language has been edited by Bioedit, a professional proofreading company.

Round 2

Reviewer 2 Report

Dear Authors,

Am glad that my review appeared to be helpful. The manuscript is now ready to be published.

Regards,